# Predictive Limitations of the Geriatric Trauma Outcome Score: A Retrospective Analysis of Mortality in Elderly Patients with Multiple Traumas and Severe Traumatic Brain Injury

**DOI:** 10.3390/diagnostics15050586

**Published:** 2025-02-28

**Authors:** Sebeom Jeon, Gil Jae Lee, Mina Lee, Kang Kook Choi, Seung Hwan Lee, Jayun Cho, Byungchul Yu

**Affiliations:** 1Department of Trauma Surgery, Gachon University Gil Medical Center, Incheon 21565, Republic of Korea; dsjeonse@gmail.com (S.J.); nonajugi@gilhospital.com (G.J.L.); jwh@gilhospital.com (M.L.); choikangkook@gilhospital.com (K.K.C.); surgeonrumi@gmail.com (S.H.L.); jayuncho@gilhospital.com (J.C.); 2Department of Traumatology, Gachon University College of Medicine, Incheon 21999, Republic of Korea

**Keywords:** geriatric trauma, traumatic brain injury, Geriatric Trauma Outcome Score, predictive accuracy, multiple traumas, in-hospital mortality, older adults

## Abstract

**Background/Objectives:** The Geriatric Trauma Outcome Score (GTOS) is used to predict in-hospital mortality in geriatric patients with trauma. However, its applicability to elderly patients with multiple traumas and severe traumatic brain injury (TBI) remains poorly understood. This study aimed to evaluate the predictive accuracy of the GTOS in elderly patients with multiple traumas and TBI and assess its performance in patients with mild and severe TBI. **Methods:** We retrospectively analyzed 1283 geriatric multiple trauma patients (aged ≥ 65 years) treated at a regional trauma center from 2019 to 2023. Patients were stratified into mild (head Abbreviated Injury Scale [AIS] ≤ 3) and severe (head AIS ≥ 4) TBI groups. GTOS values were calculated for each patient, and predicted mortality was compared with in-hospital mortality. GTOS predictive accuracy was assessed by analyzing the receiver operating characteristic curve. **Results:** Patients had a median Injury Severity Score of 18 (interquartile range: 10–25); 33.3% of patients received red blood cell transfusions within 24 h. The overall in-hospital mortality rate was 17.9%; GTOS predicted a mortality rate of 17.6% ± 0.17. The GTOS accurately predicted the in-hospital mortality in the entire cohort, achieving an Area Under the Curve (AUC) of 0.798. Predictive accuracy diminished for patients with severe TBI (AUC = 0.657), underestimating actual mortality (39.5% vs. 28.8% predicted). **Conclusions:** While the GTOS remains a useful tool for predicting in-hospital mortality in elderly patients with multiple traumas, it consistently underestimates mortality risk in those with severe TBI. Therefore, applying the GTOS in this patient subgroup warrants careful consideration.

## 1. Introduction

As the global population continues to age, there has been a corresponding increase in the number of elderly adults experiencing trauma [1,2]. The physiological conditions of older individuals differ significantly from those of younger individuals, posing unique challenges in trauma management [3,4]. It has been extensively demonstrated that trauma in older adults is associated with increased mortality rates [5,6]. This necessitates complex medical and ethical decision-making processes that often require balancing aggressive treatments and palliative care [7]. Clinicians frequently encounter situations in which they must choose between invasive interventions and the management of pain and symptoms, often leading to potential conflicts [8]. The urgency of these decisions is further compounded by time constraints, making it challenging to accurately determine the limits of treatment options. Therefore, accurate and timely prognostic indicators are urgently required to guide treatment decisions and provide prognostic information.

The Geriatric Trauma Outcome Score (GTOS) is a valuable predictive tool for mortality in older patients with trauma. This score is designed to predict in-hospital mortality in older patients who have sustained injuries and subsequently require hospitalization. The GTOS is notable for its simplicity, as it relies on three variables: age, Injury Severity Score (ISS), and the need for red blood cell (RBC) transfusion within 24 h [9,10]. Traumatic brain injury (TBI) is commonly associated with multiple trauma cases and is a leading cause of mortality in this demographic [11]. TBI is particularly severe in older patients [12]. Nevertheless, studies evaluating the reliability of the GTOS in patients with multiple traumas and severe TBI remain limited. Although tools for predicting mortality in patients with TBI exist, their complexity often limits their practical application in clinical settings.

This study aimed to evaluate the predictive accuracy of the GTOS, specifically in elderly patients with multiple traumas and TBI. This evaluation aimed to determine whether the GTOS is a reliable tool for predicting outcomes in this subgroup of elderly patients with trauma, thereby assisting in clinical decision-making and improving patient care outcomes.

## 2. Materials and Methods

### 2.1. Study Population and Data Collection

Between 2019 and 2023, a retrospective study was conducted on geriatric patients with multiple traumas aged ≥65 years who were treated at Incheon Regional Trauma Center. In Korea, a regional trauma center, the highest-level trauma center designation, is equivalent to a typical Level I trauma center. Data were sourced from the Korean Trauma Database (KTDB) at the regional trauma center. KTDB is a nationally managed, standardized database populated by trauma surgeons and trauma coordinators. Its accuracy and completeness are rigorously maintained through regular multidisciplinary meetings. The dataset included variables such as age, sex, mechanism of injury, ISS, head Abbreviated Injury Scale (AIS) score, length of hospital stay, length of intensive care unit stay, requirement for RBC transfusion within the first 24 h, and in-hospital mortality rates. We defined severe TBI as a head AIS of 4 or higher, which represents life-threatening brain injuries such as large intracranial hemorrhages and severe brain swelling [13]. This threshold is widely used in trauma research and clinical practice to identify patients at high risk of mortality and poor neurological outcomes. Patients who arrived in a state of death, died in the trauma bay, or were transferred to other facilities from the trauma bay were excluded from this study. Missing values were minimal and thus simply excluded from the final analyses. Before initiating the study, an ethics review was conducted by the Institutional Review Board, and formal approval was granted (Approval number: GAIRB2025-004).

### 2.2. GTOS

The GTOS was calculated as age + (2.5 × ISS) + 22 (if any packed RBCs were administered within the first 24 h after injury). We calculated the GTOS for all patients using these data. A logistic model for predicting mortality based on the GTOS is well established (Equation (1)). Using this logistic model, we calculated the predicted mortality rate for all patients based on their GTOS scores.(1)$$GTOS\;Predicted\;Mortality=\frac{e^{\left\{-6.9115+\left(0.03912\times GTOS\right)\right\}}}{1+e^{\left\{-6.9115+\left(0.03912\times GTOS\right)\right\}}}$$

### 2.3. Statistical Analysis

For categorical data, the chi-square test was employed, whereas continuous variables were analyzed using Student’s *t*-test. Categorical variables are presented as frequencies and percentages, whereas continuous variables are reported as means and standard deviations or medians and interquartile ranges, depending on their distribution. The GTOS was calculated for each patient using variables such as age, ISS, and the need for blood transfusion within the first 24 h. The predictions produced by the GTOS were subsequently compared with the actual in-hospital mortality outcomes observed in the patients. These predictions were analyzed and compared with the actual in-hospital mortality based on the severity of head injury, classified by head AIS scores ranging from 1 to 5. The performance of GTOS in predicting outcomes within the cohort was evaluated using receiver operating characteristic (ROC) curves, and the Area Under the Curve (AUC) was measured. The Hosmer–Lemeshow goodness-of-fit test was used to assess the model calibration. Statistical analyses were conducted using SPSS software (version 22.0; SPSS Inc., Chicago, IL, USA).

## 3. Results

A total of 5574 patients with multiple traumas visited our regional trauma center between 2019 and 2023. Among them, 1434 patients were aged ≥65 years. After excluding ineligible patients, 1283 elderly adults with multiple traumas met the inclusion criteria. These patients were categorized into two distinct groups: mild TBI (head AIS ≤ 3, *n* = 893) and severe TBI (head AIS ≥ 4, *n* = 390) (Figure 1).

Table 1 shows the clinical characteristics of both groups. Among the GTOS variables, the mean age was 74.4 ± 7.1 years for the mild TBI group and 74.7 ± 7.4 years for the severe TBI group, with no statistically significant difference (*p* = 0.497). Conversely, the ISS was significantly higher in the severe TBI group (27 (21–29)) than in the mild TBI group (14 (9–19)) (*p* < 0.001), and the proportion of RBC transfusions within 24 h was also significantly higher in the severe TBI group (37.7%) than that in the mild TBI group (30.9%) (*p* = 0.017).

For all patients (*n* = 1283), the mean GTOS was 126.9 ± 31.9 and the observed in-hospital mortality rate was 17.9% (*n* = 230), which closely aligned with the GTOS-predicted risk of 17.6%. The overall in-hospital mortality rate for all geriatric trauma patients, including those with TBI of any severity, was 17.9% (*n* = 230), similar to the model-predicted mortality rate of 17.6%. The AUC for all patients was 0.798 (95% CI: 0.770–0.825). When stratified by head AIS, the mild TBI group (*n* = 893) had a mean GTOS of 117.0 ± 27.6 and an observed in-hospital mortality of 8.4% (*n* = 75), compared with a predicted mortality of 12.8%. The AUC in this group was 0.780 (95% CI: 0.715–0.821). Conversely, the severe TBI group (*n* = 390) had a higher observed mortality of 39.5% (*n* = 154) relative to the predicted risk of 28.8%, with a mean GTOS of 149.6 ± 29.4. The AUC for severe TBI was 0.657 (95% CI: 0.603–0.710) (Table 2). When analyzing mortality rates according to head AIS thresholds (≥1 to ≥5), we found that the gap between observed and predicted mortality rates widened as the level of severity increased. Although the model prediction closely matched the observed rate for the entire cohort (17.6% vs. 17.9%), this discrepancy increased at higher head AIS levels. For instance, at head AIS ≥ 4, the observed mortality of 39.5% notably exceeded the predicted 28.8%, and among patients with head AIS ≥ 5, the gap became even more pronounced (50.9% vs. 35.3%). These findings indicate that the model systematically underestimated the mortality risk in patients with increasingly severe head injuries (Appendix A). Figure 2 illustrates the ROC curves for all elderly patients with multiple traumas (blue), mild TBI (green), and severe TBI (red). In the ROC curves, elderly patients with multiple traumas and mild TBI exhibited a level of discriminative ability comparable to that of all patients, whereas it was reduced in those with severe TBI.

Figure 3 displays the Hosmer–Lemeshow plots comparing the observed and predicted mortality rates for all patients, mild TBI, and severe TBI groups. In the mild TBI group (Figure 3B), the observed mortality rates generally followed the predicted values across most ranges, with minor discrepancies at higher predicted mortality intervals (HL = 21.77, *p* > 0.005). In cases of severe TBI (Figure 3C), the statistical significance was much higher (HL = 92.73, *p* < 0.0001). These findings suggested that the predictive model lacked adequate calibration, particularly in the severe TBI subgroup.

## 4. Discussion

This study provides notable insights into the variability in the predictive accuracy of GTOS among elderly patients with multiple traumas, particularly those with TBI. The GTOS has been widely recognized as a practical tool for predicting in-hospital mortality in geriatric trauma patients, as it incorporates three clinically relevant variables that are readily available in the early stages of hospitalization: age, ISS, and the need for RBC transfusion within the first 24 h. Previous studies have consistently demonstrated its utility, reporting overall AUC values exceeding 0.80 in general geriatric trauma populations [9,14,15]. These findings reinforce the GTOS as an effective and accessible prognostic model for trauma care in elderly patients. However, studies evaluating the applicability of the GTOS in specific subgroups, such as older patients with multiple traumas and severe TBI, remain limited. A study utilizing data from the Spanish Trauma Registry found that the predictive performance of the GTOS significantly decreased in patients with severe intracranial hypertension [16,17]. These findings suggest that the GTOS has limited predictive power for elderly trauma patients with conditions related to intracranial hypertension. Another study concluded that the GTOS demonstrated considerable accuracy in predicting in-hospital mortality among patients with TBI (AUC = 0.812), but this study was limited to patients with isolated TBI [18]. Considering that most severe TBI cases are accompanied by multiple traumas, the applicability of these results is limited.

The results of our study confirm that while the GTOS performs well in predicting in-hospital mortality in the general geriatric trauma population (AUC = 0.798), its predictive accuracy diminishes significantly as TBI severity increases. For patients with head AIS ≥ 3, the AUC decreased to 0.743, and for those with head AIS ≥ 4, it dropped further to 0.658. These findings highlight the limitations of the GTOS in accurately predicting outcomes in patients with severe TBI and emphasize the need for caution when applying the GTOS to this subgroup. The Hosmer–Lemeshow test indicated that the GTOS lacked adequate calibration for patients with severe TBI. This discrepancy between predicted and observed mortality underscores the need for further refinement or validation in this high-risk subgroup. Additional research, including prospective validation with larger and more diverse cohorts, may be warranted to enhance the model’s accuracy.

The decline in the predictive performance of the GTOS for patients with severe TBI can be attributed to several factors. First, the GTOS formula does not sufficiently account for the unique complexities associated with TBI. Although the GTOS incorporates age, ISS, and the need for early blood transfusion, it does not include neurological variables that are crucial for TBI prognosis, such as Glasgow Coma Scale scores, intracranial pressure, or the extent of cerebral edema [19]. The AIS score, which primarily evaluates anatomical severity, is limited in its ability to capture the functional and physiological effects of TBI. Patients with severe TBI often present with diverse neurological injuries that significantly influence outcomes. However, the simplified scoring system cannot fully reflect this complexity, potentially leading to an underestimation of mortality risk [20]. Second, the inclusion of RBC transfusion as a key predictor in the GTOS may not be suitable for patients with TBI. Although RBC transfusion is a reliable indicator of systemic hemorrhage, it does not accurately represent pathological processes specific to TBI, such as intracranial bleeding or localized damage [21]. In cases of severe TBI, the need for transfusion may not be directly correlated with injury severity or mortality risk, further reducing the reliability of the GTOS in this subgroup [22,23]. This mismatch suggests that the role of transfusion as a predictive variable should be re-evaluated, particularly in patients with isolated or predominant head injuries. Third, the GTOS does not account for complications that significantly impact outcomes in elderly patients with multiple traumas. This omission is particularly relevant for patients with TBI as they are more likely to develop long-term complications such as infections, thromboembolic events, or neurocognitive decline, which contribute to increased mortality [24,25,26]. Additionally, elderly patients often have preexisting comorbidities that exacerbate the effects of trauma and increase their vulnerability to complications [27,28]. The observed mortality rate in our cohort (39.5%) was markedly higher than the GTOS-predicted mortality rate (28.8%), which may be partially explained by systemic complications, delayed specialized care, and unique regional factors not captured by GTOS. This discrepancy suggests that factors beyond the GTOS variables may contribute to the increased mortality in this group. The inability of the GTOS to capture these factors may lead to an underestimation of mortality risk and reduced predictive accuracy for patients with severe TBI.

These findings have significant clinical implications. First, the accurate prediction of mortality in elderly patients with multiple traumas is critical for guiding treatment decisions, resource allocation, and discussions with families regarding prognosis and treatment goals [29,30,31]. The reduced predictive accuracy of the GTOS in patients with severe TBI suggests that clinicians should exercise caution when relying solely on the GTOS for prognostication in this subgroup. The GTOS is suggested to be used in time- and resource-limited emergencies and clinical settings. Additionally, this study emphasizes the need for ongoing evaluation and validation of prognostic tools across diverse patient populations. Trauma care is a dynamic field, and predictive models must be regularly updated and refined to reflect current clinical practices and patient demographics. Additionally, exploring modifications to the GTOS, such as incorporating neurological variables, biomarkers for brain injury, or advanced imaging findings, could enhance its predictive accuracy and utility in elderly patients with trauma and TBI. For instance, we attempted a method to improve the GTOS in geriatric trauma patients with TBI by integrating the GCS, which is relatively easy to obtain in the early stage. Specifically, we refined the GTOS by subtracting the GCS from 15, and then—using a grid search to identify optimal coefficients—multiplying (15 – GCS) by 8 before adding it to the GTOS. This revised GTOS showed improved AUC performance (Appendix A). The second important clinical implication relates to the potential role of severe TBI as a marker of limited benefit from resuscitative efforts in geriatric trauma patients with hemorrhagic shock. While some recent studies emphasize the importance of early futility assessment in severe TBI [32,33,34], our data are not sufficient to conclude that the presence of severe TBI alone justifies discontinuing resuscitation efforts. Specifically, we did not perform time-to-death analyses or measure long-term neurological outcomes, which would provide more direct evidence. Nonetheless, the significantly higher observed in-hospital mortality in geriatric patients with multiple traumas and severe TBI, compared to the mortality risk predicted by GTOS, suggests that severe TBI may be associated with poorer outcomes than anticipated. This finding underscores the need for further research to clarify whether—and under what conditions—resuscitation may be futile in the context of hemorrhagic geriatric trauma. If early-stage mortality can be more clearly defined and robust futility criteria established, it could ultimately help optimize resource allocation, including the judicious use of blood products [35,36].

This study had a few limitations. First, it used data from a single regional trauma center, limiting the generalizability of the findings to other institutions or regions where differences in trauma care systems, patient characteristics, or treatment protocols may yield different outcomes. Second, the retrospective design introduces potential biases and missing data, as some variables may have been inaccurately recorded or key information, such as causes of mortality, may have been omitted, and by excluding patients who died immediately upon arrival or were transferred, the most critically injured group may have been omitted from the study, potentially leading to an overestimation of the GTOS’s predictive accuracy in the overall cohort. Third, the exclusion of patients who were dead on arrival or transferred to other hospitals from the emergency department may have introduced survivorship bias, potentially altering the predictive performance of GTOS if these patient groups had been included. Fourth, we did not perform analyses that accounted for comorbidities. The underlying conditions of geriatric trauma patients may affect the predictive accuracy of the GTOS. Finally, our study did not propose a modified GTOS that incorporates neurological variables to potentially improve predictive accuracy in patients with severe TBI. We attempted to improve the GTOS by integrating the GCS as one possible approach. However, because these findings are based on a limited dataset from our institution, they may not be generalizable, and it is important to note that the GCS cannot be measured in some patients. This study emphasizes the need for continuous evaluation and validation of prognostic tools across diverse patient populations. In future research, it may be necessary to integrate neurological indicators. Given that GTOS relies on readily calculable variables, incorporating early assessable neurological measures such as the GCS is essential for its refinement. Moreover, large scale multicenter studies should be conducted to validate these modifications, with the goal of enhancing GTOS by improving prognostic accuracy for high-risk subgroups.

## 5. Conclusions

The GTOS is an effective tool for predicting in-hospital mortality in elderly patients with trauma; however, its accuracy decreases in elderly patients with multiple traumas with severe TBI. Because the data in this study were derived from a single trauma center, the findings may not be fully generalizable and should therefore be interpreted in the context of the study design, population, and limitations. To improve predictive reliability for this subgroup, adjustments to the GTOS, such as incorporating neurological variables, are necessary. Future validation studies are essential to enhance the utility of the GTOS across diverse trauma populations.

## Figures and Tables

**Figure 1 diagnostics-15-00586-f001:**
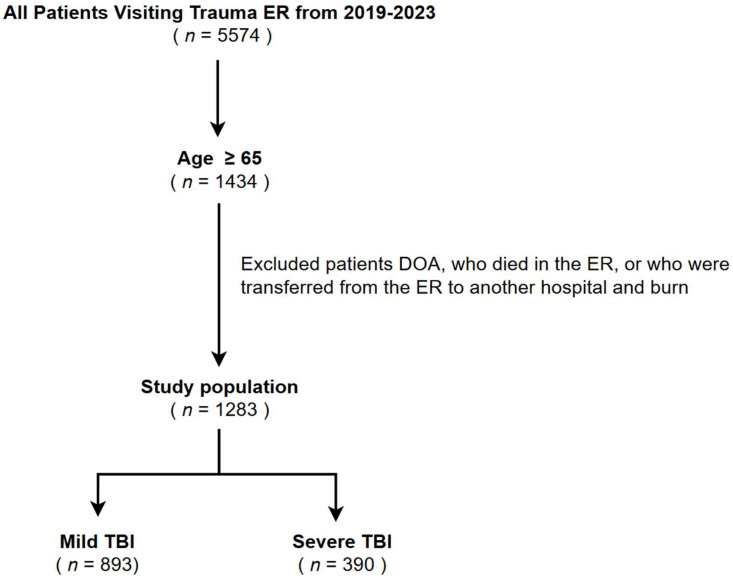
Flow diagram of patient selection for mild and severe TBI among elderly patients with multiple traumas from 2019 to 2023.

**Figure 2 diagnostics-15-00586-f002:**
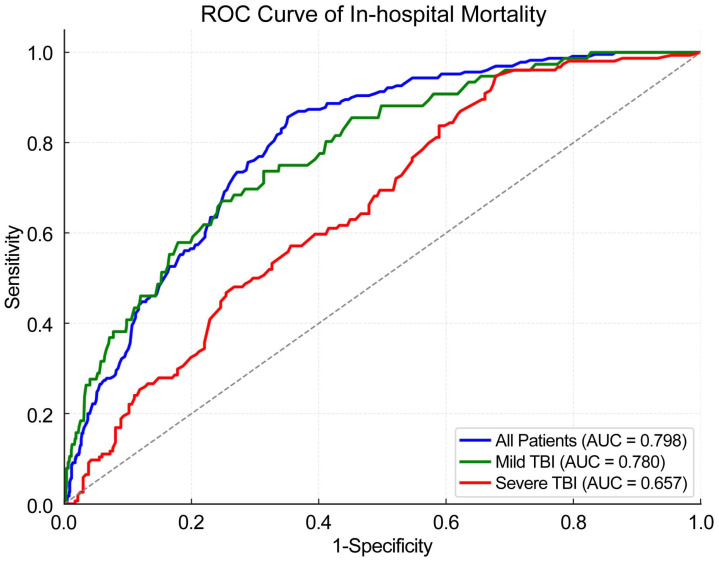
Receiver operating characteristic (ROC) curve for in-hospital mortality prediction using GTOS.

**Figure 3 diagnostics-15-00586-f003:**
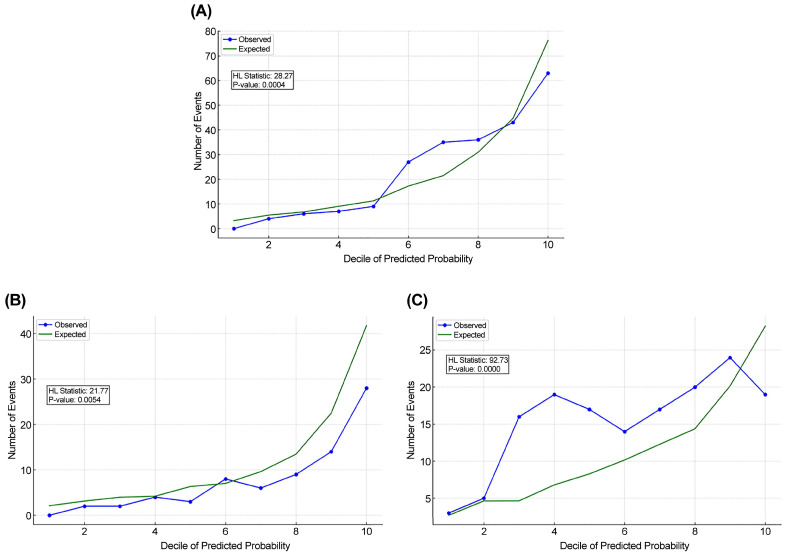
Hosmer–Lemeshow goodness-of-fit plots for predicted mortality in elderly patients with TBI and multiple traumas, including all patients (**A**), those with mild TBI (**B**), and those with severe TBI (**C**).

**Table 1 diagnostics-15-00586-t001:** Clinical characteristics and outcomes of patients with mild and severe TBI and multiple traumas.

Characteristics	Mild TBI (*n* = 893)	Severe TBI (*n* = 390)	*p*-Value
Age, years, mean ± SD	74.4 ± 7.1	74.7 ± 7.4	0.497
Male, *n* (%)	586 (65.6)	290 (74.4)	0.002
Mechanism, *n* (%)			<0.001
MVC	121 (13.5)	15 (3.8)	
MCC	82 (9.2)	25 (6.4)	
Bicycle	44 (4.9)	28 (7.2)	
AVP	162 (18.1)	55 (14.1)	
Fall	215 (24.1)	84 (21.5)	
Struck by object	39 (4.4)	14 (3.6)	
Slip	153 (17.1)	143 (36.7)	
Penetrating	53 (5.9)	0 (0.0)	
Unknown	15 (1.7)	24 (6.2)	
Others	9 (1.0)	2 (0.5)	
Vital signs at ED			
SBP, mm Hg, mean ± SD	140.3 ± 36.3	155.4 ± 39.6	<0.001
HR, bpm, mean ± SD	86.2 ± 19.4	87.9 ± 22.3	0.177
RR, /min, mean ± SD	20.8 ± 4.2	20.0 ± 4.1	0.001
GCS, median (IQR)	14 (14–15)	9 (3–14)	<0.001
Hospital visit type, *n* (%)			0.003
Direct	443 (49.6)	229 (58.7)	
Transfer	450 (50.4)	161 (41.3)	
Injury Severity Parameter			
ISS, median (IQR)	14 (9–19)	27 (21–29)	<0.001
ISS > 15, *n* (%)	355 (39.8)	390 (100.0)	<0.001
Hospital course, days mean ± SD			
Hospital LOS	19.0 ± 22.3	24.8 ± 28.6	<0.001
ICU LOS	4.1 ± 9.8	10.7 ± 17.1	<0.001
RBC within 24 h, *n* (%)	276 (30.9)	147 (37.7)	0.017
GTOS, mean ± SD	117.0 ± 27.6	149.6 ± 29.4	<0.001
In-hospital mortality, *n* (%)	75 (8.4)	154 (39.5)	<0.001

Abbreviations: SD, standard deviation; MVC, Motor Vehicle Crash; MCC, Motorcycle Crash; AVP, Automobile Versus Pedestrian; ED, emergency department; SBP, systolic blood pressure; HR, heart rate; GCS, Glasgow Coma Scale; ISS, Injury Severity Score; ICU, intensive care unit; LOS, length of stay; RBCs, Red Blood Cells; GTOS, Geriatric Trauma Outcome Score.

**Table 2 diagnostics-15-00586-t002:** Observed and predicted mortality using the Geriatric Trauma Outcome Score (GTOS) and model performance in elderly patients with mild and severe TBI and multiple traumas.

Group	*N*	GTOS	Observed In-Hospital Mortality, % (*n*)	GTOS Predicted Risk of Mortality, (%)	AUC	95% CI
All Patients	1283	126.9 ± 31.9	17.9 (230)	17.6	0.798	0.770–0.825
Mild TBI	893	117.0 ± 27.6	8.4 (75)	12.8	0.780	0.715–0.821
Severe TBI	390	149.6 ± 29.4	39.5 (154)	28.8	0.657	0.603–0.710

Abbreviations: GTOS, Geriatric Trauma Outcome Score; AUC, Area Under the Curve; CI, Confidence Interval; TBI, traumatic brain injury.

## Data Availability

The data presented in this study are available on request from the corresponding author. The data are not publicly available due to privacy or ethical restrictions.

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
