# Peer review of "Predictive Limitations of the Geriatric Trauma Outcome Score: A Retrospective Analysis of Mortality in Elderly Patients with Multiple Traumas and Severe Traumatic Brain Injury"

_diagnostics, 2025, doi:10.3390/diagnostics15050586_

Round 1

Reviewer 1 Report

Comments and Suggestions for Authors

Review Report: Diagnostics Review Request

Title: Evaluation of the Geriatric Trauma Outcome Score in Elderly Patients with Multiple Traumas and Traumatic Brain Injury

The study assesses the predictive accuracy of the Geriatric Trauma Outcome Score (GTOS) in elderly patients with multiple traumas, particularly those with traumatic brain injury (TBI). The research involves a retrospective analysis of 1,283 patients aged 65 and above, treated at a regional trauma center from 2019 to 2023. The study evaluates the GTOS’s ability to predict in-hospital mortality and its performance in patients with mild and severe TBI. Findings indicate that while the GTOS is an effective tool for general trauma mortality prediction, it systematically underestimates mortality in severe TBI cases.

The study addresses a significant gap in trauma prognosis by evaluating the effectiveness of an established scoring system in a critical patient subgroup. Including 1,283 patients enhances the statistical power and reliability of the findings. The findings highlight the need for caution when using GTOS in patients with severe TBI and suggest potential refinements to improve prognostic accuracy.

I have the following comments for the authors: 

1. The authors should discuss possible modifications to GTOS that could enhance its accuracy, such as incorporating neurological indicators like the Glasgow Coma Scale or biomarkers.

2. Given this limitation, the authors should explore alternative scoring models or adjustments to GTOS to improve its applicability to this specific patient population.

3. Some sections could benefit from clearer definitions of key terms, particularly the classification criteria for mild versus severe TBI.

4. The methodology section should explicitly state whether ISS and AIS scores were assigned by independent reviewers or extracted from medical records.

5. More information on how missing data were handled would strengthen the study’s transparency.

6. The justification for using specific cutoffs for mild versus severe TBI should be elaborated.

7. Future Directions: The study rightly emphasizes the need for ongoing validation of predictive models but does not propose a modified version of GTOS. The authors should provide recommendations for future studies, including prospective validation of a revised GTOS incorporating additional neurological parameters.

Overall, the study provides valuable insights into the limitations of GTOS for elderly trauma patients with TBI. While the research is well-structured and clinically relevant, it highlights the need for refinements in predictive models to account for neurological variables and other factors affecting TBI prognosis. Minor improvements in clarity and methodological details would enhance readability and reproducibility.

Author Response

Comment 1: The authors should discuss possible modifications to GTOS that could enhance its accuracy, such as incorporating neurological indicators like the Glasgow Coma Scale or biomarkers.

Response 1: Thank you very much for your important and insightful comments. Our research team also had extensive discussions during the study regarding the issues you raised. We determined that integrating neurological variables would be necessary to improve the GTOS, and we set a criterion that any neurological variable included must be easy to measure at an early stage. Therefore, we attempted to refine the GTOS by incorporating the Glasgow Coma Scale (GCS). Specifically, we tried adding (15−GCS) to the existing GTOS and used a grid search method to find the appropriate coefficient to multiply with (15−GCS). We then calculated the AUC for the revised GTOS and observed improved performance. However, our approach had several limitations. First, it was based solely on data from our own institution. Second, because the GCS is not always measurable in certain patients, this method cannot be universally applied. Consequently, we concluded that our findings could not be generalized, and we decided not to include this analysis in the main manuscript. We have added the relevant content in lines 232–237 and 261–263. Following your feedback, we revisited the analysis and have decided to include it in the supplementary material.

Comment 2: Given this limitation, the authors should explore alternative scoring models or adjustments to GTOS to improve its applicability to this specific patient population.

Response 2: As mentioned in Response 1, we have incorporated this information accordingly. Once again, we sincerely appreciate your valuable insights.

Comment 3: Some sections could benefit from clearer definitions of key terms, particularly the classification criteria for mild versus severe TBI.

Response 3: We appreciate the reviewer’s suggestion regarding clearer definitions. In the revised manuscript, we have clarified in the Methods section that a Head AIS score of 4 or above was used to define severe TBI. This content has been added to the Methods section, lines 75–76.

Comment 4: The methodology section should explicitly state whether ISS and AIS scores were assigned by independent reviewers or extracted from medical records.

Response 4: In the revised manuscript, we specify that the Injury Severity Score (ISS) and the Abbreviated Injury Scale (AIS) were obtained from standardized trauma registry data, which are independently reviewed and validated by dedicated trauma registrars at our center. We have added a statement in line 70-72 clarifying that the data are managed by experienced registrars, rather than the authors, to ensure objectivity and reliability.

Comment 5: More information on how missing data were handled would strengthen the study’s transparency.

Response 5: We appreciate the reviewer’s inquiry regarding our handling of missing data. In our dataset, missing values were minimal and therefore excluded from the final analysis. As such, we did not perform any advanced imputation or correction methods. We acknowledge that more sophisticated techniques (e.g., multiple imputation) could be considered in cases where missing data are more substantial. However, in our study, the low volume of missing values allowed us to maintain adequate statistical power and minimize bias through complete-case analysis. We have clarified this point in the Methods section in line 78-79 to enhance the transparency and reproducibility of our findings.

Comment 6: The justification for using specific cutoffs for mild versus severe TBI should be elaborated.

Response 6: We used the Abbreviated Injury Scale (AIS) for head injuries to categorize mild TBI (AIS ≤3) and severe TBI (AIS ≥4). While there is no universally accepted standard for TBI classification, these cutoffs were chosen to align with commonly cited AIS-based severity thresholds and reflect meaningful clinical distinctions in head trauma severity. We acknowledge that other classification schemes (e.g., Glasgow Coma Scale) exist, but our AIS-based approach facilitates consistency with certain trauma registries and prior research.

Comment 7: Future Directions: The study rightly emphasizes the need for ongoing validation of predictive models but does not propose a modified version of GTOS. The authors should provide recommendations for future studies, including prospective validation of a revised GTOS incorporating additional neurological parameters.

Response 7: We appreciate the reviewer’s insights into the future direction of GTOS. In the Discussion, we highlighted the potential value of integrating additional neurological parameters to refine GTOS in elderly patients with multiple traumas accompanied by TBI. Given that GTOS can be readily calculated in the early stage, incorporating early-assessable neurological variables is critical to its enhancement. We plan to investigate these refinements through a prospective, large-scale, multicenter validation study, which we hope will confirm whether a revised GTOS can provide more accurate prognostic utility for this high-risk population. This content has been revised, added, and reorganized in the Discussion section, lines 264–269.

Reviewer 2 Report

Comments and Suggestions for Authors

It is a privilege to review the manuscript submitted to Diagnostics  entitled:

Validation of the Geriatric Trauma Outcome Score for Predicting In-hospital Mortality in Elderly Patients with Multiple 3 Traumas and Traumatic Brain Injury.

In my review below I have a section about transparency where I stated that  the authors did not need to refer to my own two papers which I had coauthored. I have given the authors many other papers which address the same concerns but with much less detail than the two papers listed below. I also have also referred to one section from one of the two papers which would help the authors refine their message. I have specifically stated that the authors merely review the section but need not refer to the paper; I have therefor reiterated in the  review  below clearly stating that the authors may not refer to my own papers but I also feel that the authors would benefit by reading the papers and then they could adjust their message as I have suggested to make their manuscript more relevant for the bedside prediction of mortality for severely bleeding trauma patients with head injury.

I have marked the two papers below which the authors may not cite, but which will be very useful for expanding the breath of their message to include predictors of early death for severely injured trauma patients with traumatic brain

Al-Fadhl MD, Karam MN, Chen J, Zackariya SK, Lain MC, Bales JR, Higgins AB, Laing JT, Wang HS, Andrews MG, Thomas AV, Smith L, Fox MD, Zackariya SK, Thomas SJ, Tincher AM, Al-Fadhl HD, Weston M, Marsh PL, Khan HA, Thomas EJ, Miller JB, Bailey JA, Koenig JJ, Waxman DA, Srikureja D, Fulkerson DH, Fox S, Bingaman G, Zimmer DF, Thompson MA, Bunch CM, Walsh MM; Futile Indicators for Stopping Transfusion in Trauma (FISTT) Collaborative Group. Traumatic Brain Injury as an Independent Predictor of Futility in the Early Resuscitation of Patients in Hemorrhagic Shock. J Clin Med. 2024 Jul 3;13(13):3915. doi: 10.3390/jcm13133915. PMID: 38999481; PMCID: PMC11242176.

Walsh MM, Fox MD, Moore EE, Johnson JL, Bunch CM, Miller JB, Lopez-Plaza I, Brancamp RL, Waxman DA, Thomas SG, Fulkerson DH, Thomas EJ, Khan HA, Zackariya SK, Al-Fadhl MD, Zackariya SK, Thomas SJ, Aboukhaled MW, The Futile Indicators For Stopping Transfusion In Trauma FISTT Collaborative Group. Markers of Futile Resuscitation in Traumatic Hemorrhage: A Review of the Evidence and a Proposal for Futility Time-Outs during Massive Transfusion. J Clin Med. 2024 Aug 9;13(16):4684. doi: 10.3390/jcm13164684. PMID: 39200824; PMCID: PMC11355875.

The authors have brought to light important evidence appreciation for the importance of better defining mortality in patients with severe trauma when using various predictors of mortality and injury severity particularly in elderly patients. Their careful and straightforward  study is an important reminder that the elderly are predisposed to worse outcomes when they are subjected to multiple trauma particularly if associated with traumatic brain injury (TBI). In their study they evaluate the ability of the Geriatric Trauma Outcome Score (GTOS) to predict in-hospital mortality in geriatric patients with trauma. Their study is aimed to evaluate the predictive accuracy of the GTOS in elderly patients with multiple trauma and TBI and assess its performance in patients with  mild and severe TBI.

 They  retrospectively analyzed 1,283 geriatric multiple trauma.  Patients were stratified into mild Abbreviated Injury Scores (AIS) <3 and severe AIS > 4  TBI groups . GTOS values were calculated for each patient and predicted mortality was compared with in-hospital mortality. GTOS predictive accuracy was assessed by analyzing receiver operating characteristic curve.

Patients had a median Injury Severity Score (ISS) of 18 (interquartile range: 10–25); 33.3% of patients received red blood cell transfusions within 24 h. The overall in-hospital mortality rate was 17.9%; GTOS predicted a mortality rate of 17.6% ± 0.17. However, predictive accuracy diminished for patients with severe TBI, underestimating actual mortality (39.5% vs. 28.8% predicted). They conclude that the GTOS remains a useful tool for predicting in-hospital mortality in elderly patients with multiple traumas, it consistently underestimates mortality risk in those with severe TBI. Therefore, applying the  GTOS in this patient subgroup warrants careful consideration.

I congratulate the authors for their study and the conclusions at which they have arrived.

I would request that the authors consider amending  their draft in the following ways.

The authors describe the ability of the GTOS to predict mortality at 24 hours which is the standard for the GTOS score. However, to be clinically useful, I propose that it would be useful to briefly suggest that there are current attempts to define mortality even earlier so as to protect the dwindling blood supply which has beset major trauma centers in the United States, for example. This addition might make this manuscript much more useful for the practicing traumatologist. I will outline my rationale below. The authors are free to accept my suggestion of adding this reference to the early identification of patients who will not survive major trauma who also have head injury or add a brief section about the need to refine a prediction tool for identifying  trauma patients who will not survive major trauma and where a score like the GTOS, but available in the early stages of resuscitation might be useful.

Due to the prohibition of referencing  a reviewer’s own work, I am mentioning two papers which I have coauthored recently which the authors may not  cite, but they can find information that might help them add a brief section about the urgency of defining mortality earlier than the 24 hour mortality as in the GTOS which is useful for epidemiologic analysis but practical not for the early moments of trauma resuscitation which is when scarce blood products are used. I believe that a brief mentioning of the urgency of defining with greater detail, mortality for elderly bleeding trauma patients who also have head injury is a worthy addition that will make this manuscript much more relevant and practical to the practicing traumatologist.

  1. The author’s manuscript was most likely written and submitted during the latter part of 2024 before which a few important papers had been published which support the authors conclusions. Two of these papers I have co-authors, and the authors are not allowed to cite these papers. Specifically, Al Fadhl et al., Walsh et al. and Bhogadi et al. have clearly described early markers for defining futility for severely bleeding trauma patients who also have associated head injury. In those three papers, the presence of severe brain injury markedly worsens the prognosis such that the presence of associated severe TBI is an independent factor which weighs heavily on the decision to terminate futile renunciation early in the course of trauma resuscitation for severely bleeding trauma with severe TBI and extracranial injury. The Bhogadi et al. paper may be cited.

These papers are noted below.

Al-Fadhl MD, Karam MN, Chen J, Zackariya SK, Lain MC, Bales JR, Higgins AB, Laing JT, Wang HS, Andrews MG, Thomas AV, Smith L, Fox MD, Zackariya SK, Thomas SJ, Tincher AM, Al-Fadhl HD, Weston M, Marsh PL, Khan HA, Thomas EJ, Miller JB, Bailey JA, Koenig JJ, Waxman DA, Srikureja D, Fulkerson DH, Fox S, Bingaman G, Zimmer DF, Thompson MA, Bunch CM, Walsh MM; Futile Indicators for Stopping Transfusion in Trauma (FISTT) Collaborative Group. Traumatic Brain Injury as an Independent Predictor of Futility in the Early Resuscitation of Patients in Hemorrhagic Shock. J Clin Med. 2024 Jul 3;13(13):3915. doi: 10.3390/jcm13133915. PMID: 38999481; PMCID: PMC11242176.

Walsh MM, Fox MD, Moore EE, Johnson JL, Bunch CM, Miller JB, Lopez-Plaza I, Brancamp RL, Waxman DA, Thomas SG, Fulkerson DH, Thomas EJ, Khan HA, Zackariya SK, Al-Fadhl MD, Zackariya SK, Thomas SJ, Aboukhaled MW, The Futile Indicators For Stopping Transfusion In Trauma FISTT Collaborative Group. Markers of Futile Resuscitation in Traumatic Hemorrhage: A Review of the Evidence and a Proposal for Futility Time-Outs during Massive Transfusion. J Clin Med. 2024 Aug 9;13(16):4684. doi: 10.3390/jcm13164684. PMID: 39200824; PMCID: PMC11355875.

Bhogadi SK, Ditillo M, Khurshid MH, Stewart C, Hejazi O, Spencer AL, Anand T, Nelson A, Magnotti LJ, Joseph B. Development and Validation of Futility of Resuscitation Measure in Older Adult Trauma Patients. J Surg Res. 2024 Sep;301:591-598. doi: 10.1016/j.jss.2024.07.019. Epub 2024 Aug 1. PMID: 39094517. ( This paper may be cited since I am not a coauthor. The senior author of this paper,  Bellal Joseph, is a leading proponent of the GTOS and he is searching for new bedside predictors to refine the GTOS to identify futility in trauma resuscitation much earlier in order to preserve scarce resources.

  1. A motivation for the above mentioned papers is that there are no current guidelines which provide clear direction for defining futility for severely bleeding trauma patients with or without associated TBI. The need for such guidelines which can be applied earlier in the course of trauma resuscitation has been brought to the forefront by the dwindling supply of blood products available for trauma resuscitation for these patients. The cause of the significant blood shortage in the United states, for example, is, among other causes, the increased use of hemostatic resuscitation whereby, patents are resuscitated with large quantities of blood products in ratios of 1/1/1/ packed red cells, plasma and platelets with increasing use of whole blood. The unintended consequence of this strategy is a shortage blood products for future trauma resuscitation as well as for other emergencies. Recent literature has documented the great strain in blood banks which are now faced with shortages of vital blood components during peak periods of trauma.

I would therefore suggest that the authors add the paper by  described above. The first two papers cannot be cited.

Bhogadi SK, Ditillo M, Khurshid MH, Stewart C, Hejazi O, Spencer AL, Anand T, Nelson A, Magnotti LJ, Joseph B. Development and Validation of Futility of Resuscitation Measure in Older Adult Trauma Patients. J Surg Res. 2024 Sep;301:591-598. doi: 10.1016/j.jss.2024.07.019. Epub 2024 Aug 1. PMID: 39094517. ( This paper may be cited since I am not a coauthor. The senior author of this paper,  Bellal Joseph, is a leading proponent of the GTOS and he is searching for new bedside predictors to refine the GTOS to identify futility in trauma resuscitation much earlier in order to preserve scarce resources.

to their bibliography and also highlight the importance of using the presence of TBI as an important marker for futility which, therefore, will assist traumatologists in identifying those patients with extracranial hemorrhage who will not survive resuscitative efforts.

  1. In addition, a passing reference to the need to more clearly define mortality early during trauma resuscitation because of the shortage of blood products would make this paper more relevant to the traumatologist who will need to use tools like the GTOS to predict death earlier during the course of trauma resuscitation in order to spare blood products for those younger patients without TBI who will benefit from continued resuscitation. These papers are the following.

Van Gent JM, Clements TW, Lubkin DT, Wade CE, Cardenas JC, Kao LS, Cotton BA. Predicting Futility in Severely Injured Patients: Using Arrival Lab Values and Physiology to Support Evidence-Based Resource Stewardship. J Am Coll Surg. 2023 Apr 1;236(4):874-880. doi: 10.1097/XCS.0000000000000563. Epub 2023 Jan 18. PMID: 36728085.

Clements TW, Van Gent JM, Lubkin DE, Wandling MW, Meyer DE, Moore LJ, Cotton BA. The reports of my death are greatly exaggerated: An evaluation of futility cut points in massive transfusion. J Trauma Acute Care Surg. 2023 Nov 1;95(5):685-690. doi: 10.1097/TA.0000000000003980. Epub 2023 May 1. PMID: 37125814.

Loudon AM, Rushing AP, Hue JJ, Ziemak A, Sarode AL, Moorman ML. When is enough enough? Odds of survival by unit transfused. J Trauma Acute Care Surg. 2023 Feb 1;94(2):205-211. doi: 10.1097/TA.0000000000003835. Epub 2022 Nov 10. PMID: 36694331.

Saillant NN, Kornblith LZ, Moore H, Barrett C, Schreiber MA, Cotton BA, Neal MD, Makar R, Cap AP. The National Blood Shortage-An Impetus for Change. Ann Surg. 2022 Apr 1;275(4):641-643. doi: 10.1097/SLA.0000000000005393. PMID: 35081570; PMCID: PMC9055632.

Mladinov D, Frank SM. Massive transfusion and severe blood shortages: establishing and implementing predictors of futility. Br J Anaesth. 2022 Feb;128(2):e71-e74. doi: 10.1016/j.bja.2021.10.013. Epub 2021 Nov 15. PMID: 34794769.

  1. Again, as mentioned above; in order to display transparency, this reviewer is a coauthor for the first two papers. The authors may not use these two papers because of the potential conflict of interest since I have suggested these papers. However, these two papers represent the most complete reviews regarding the effect of TBI on the likelihood of survival for trauma patients in severe shock who are receiving large quantities blood products. There are other less current and comprehensive papers which address the same concerns which I will add below.

Huang CY, Yen YH, Tsai CH, Hsu SY, Tsai PL, Hsieh CH. Geriatric Trauma Outcome Score as a Mortality Predictor in Isolated Moderate to Severe Traumatic Brain Injury: A Single-Center Retrospective Study. Healthcare (Basel). 2024 Aug 22;12(16):1680. doi: 10.3390/healthcare12161680. PMID: 39201238; PMCID: PMC11353928. (Very recent publication and cited as reference 17 already.)

Saillant NN, Kornblith LZ, Moore H, Barrett C, Schreiber MA, Cotton BA, Neal MD, Makar R, Cap AP. The National Blood Shortage-An Impetus for Change. Ann Surg. 2022 Apr 1;275(4):641-643. doi: 10.1097/SLA.0000000000005393. PMID: 35081570; PMCID: PMC9055632.

L'Huillier JC, Hua S, Logghe HJ, Yu J, Myneni AA, Noyes K, Guo WA. Transfusion futility thresholds and mortality in geriatric trauma: Does frailty matter? Am J Surg. 2024 Feb;228:113-121. doi: 10.1016/j.amjsurg.2023.08.020. Epub 2023 Aug 25. PMID: 37684168.

Mladinov D, Frank SM. Massive transfusion and severe blood shortages: establishing and implementing predictors of futility. Br J Anaesth. 2022 Feb;128(2):e71-e74. doi: 10.1016/j.bja.2021.10.013. Epub 2021 Nov 15. PMID: 34794769.

Maegele M, Schöchl H, Menovsky T, Maréchal H, Marklund N, Buki A, Stanworth S. Coagulopathy and haemorrhagic progression in traumatic brain injury: advances in mechanisms, diagnosis, and management. Lancet Neurol. 2017 Aug;16(8):630-647. doi: 10.1016/S1474-4422(17)30197-7. Epub 2017 Jul 11. PMID: 28721927.

Park J, Lee Y. Predicting Mortality of Korean Geriatric Trauma Patients: A Comparison between Geriatric Trauma Outcome Score and Trauma and Injury Severity Score. Yonsei Med J. 2022 Jan;63(1):88-94. doi: 10.3349/ymj.2022.63.1.88. PMID: 34913288; PMCID: PMC8688368. ( already cited as reference 13)

Benhamed A, Emond M, Mercier E, Heidet M, Gauss T, Saint-Supery P, Yadav K, David JS, Claustre C, Tazarourte K. Accuracy of a Prehospital Triage Protocol in Predicting In-Hospital Mortality and Severe Trauma Cases among Older Adults. Int J Environ Res Public Health. 2023 Jan 20;20(3):1975. doi: 10.3390/ijerph20031975. PMID: 36767343; PMCID: PMC9916137.

Raj R, Wennervirta JM, Tjerkaski J, Luoto TM, Posti JP, Nelson DW, Takala R, Bendel S, Thelin EP, Luostarinen T, Korja M. Dynamic prediction of mortality after traumatic brain injury using a machine learning algorithm. NPJ Digit Med. 2022 Jul 18;5(1):96. doi: 10.1038/s41746-022-00652-3. PMID: 35851612; PMCID: PMC9293936.

Zakrison TL, Essig R, Polcari A, McKinley W, Arnold D, Beyene R, Wilson K, Rogers S Jr, Matthews JB, Millis JM, Angelos P, O'Connor M, Mansour A, Goldenberg F, Spiegel T, Horowitz P, Das P, Slidell M, Chokshi N, Okeke I, Barth R, Wilkins HE 3rd, Kass-Hout T, Lazaridis C. Review Paper on Penetrating Brain Injury: Ethical Quandaries in the Trauma Bay and Beyond. Ann Surg. 2023 Jan 1;277(1):66-72. doi: 10.1097/SLA.0000000000005608. Epub 2022 Aug 23. PMID: 35997268; PMCID: PMC9762724.

  1. Finally, the authors state on pages 6 and 7 lines 168-173 that: “The GTOS has been widely recognized as a practical tool for predicting in-hospital mortality 167 in geriatric trauma patients, integrating three simple yet clinically relevant factors: age, 168 ISS, and the need for RBC transfusion within the first 24 h.” The reason for my enthusiasm for this manuscript is that the paper confirms our position and that of others recently in the literature as mentioned above that more accurate indicators of early death are needed to better predict those patient who will certainly die during resuscitation of severely bleeding trauma patients in order to preserve scare blood components. This paper correctly states that the severity of TBI in elderly trauma patients is an independent predictor of mortality. However, the calculation of the GTOS is not “simple” mentioned by the authors and cannot be dome at the bedside in the Emergency Department. In the paper cited above. The authors cannot cite the Al Fadhl paper, but I am copying a section from that paper which clearly analyzes why the GTOS, and the frailty index are not suitable toll of the early identification of the bleeding trauma patient who will not survive the first hours of resuscitation which is when a large preponderance of blood products and resources are used. I believe that the section in italics below will help the authors refine their message to refer to the urgent need for prediction toll which can predicts early death in bleeding trauma patients with head injury.

Al-Fadhl MD, Karam MN, Chen J, Zackariya SK, Lain MC, Bales JR, Higgins AB, Laing JT, Wang HS, Andrews MG, Thomas AV, Smith L, Fox MD, Zackariya SK, Thomas SJ, Tincher AM, Al-Fadhl HD, Weston M, Marsh PL, Khan HA, Thomas EJ, Miller JB, Bailey JA, Koenig JJ, Waxman DA, Srikureja D, Fulkerson DH, Fox S, Bingaman G, Zimmer DF, Thompson MA, Bunch CM, Walsh MM; Futile Indicators for Stopping Transfusion in Trauma (FISTT) Collaborative Group. Traumatic Brain Injury as an Independent Predictor of Futility in the Early Resuscitation of Patients in Hemorrhagic Shock. J Clin Med. 2024 Jul 3;13(13):3915. doi: 10.3390/jcm13133915. PMID: 38999481; PMCID: PMC11242176. state:  The parameter of age is considered particularly significant in the foundation studies that define the severity of TBI, although the effect of the increasing severity of frailty on the outcomes is more important and can be added to the list of predictors of mortality for the geriatric trauma patient, such as the Geriatric Trauma Outcome Score (GTOS) and the Brain Injury Guidelines (BIG), which require the calculation of a post-admission Injury Severity Score (ISS) [71–81]. In the most recent studies to identify predictors of FR in patients with TBI, the accurate scoring of a “frailty index” in trauma patients can prevent unnecessary interventions in those with a high risk of dying while not withholding intervention in those who may find it beneficial [79,80]. However, the frailty index requires the calculation of parameters that cannot be calculated at the bedside within the first few hours of arrival; therefore, there is an impetus to find triage tools for patients with TBI that can predict early mortality [32,74,76,79,82–88].

 The GTOS is not simple and cannot be easily calculated at the bedside in the ED however, this paper is an important step toward the development of bedside criteria which can accurately predict futility in severely bleeding trauma patients.

The formula for calculation of the GTOS is noted below to confirm that the score is not always simple and cannot be calculated at the bedside within the first few hours of resuscitation. This point is being made because the search for more accurate indicators of certain death for severely bleeding trauma patients will include the assessment of age and the presence of severe TBI as the first marker of futility during the earliest periods resuscitation ad will rely heavily on this type of research that the authors have skillfully presented in their submitted manuscript to Diagnostics.

 The calculation involves two equations in order to show predictive value as noted by:

Zhao FZ, Wolf SE, Nakonezny PA, Minhajuddin A, Rhodes RL, Paulk ME, Phelan HA. Estimating Geriatric Mortality after Injury Using Age, Injury Severity, and Performance of a Transfusion: The Geriatric Trauma Outcome Score. J Palliat Med. 2015 Aug;18(8):677-81. doi: 10.1089/jpm.2015.0027. Epub 2015 May 14. PMID: 25974408; PMCID: PMC4522950.

GTO score¼ Age þ (2:5 x ISS)þ22 (if given PRBCs) [Equation 1]

GTO logistic model¼ e 6:9115þ(0:03912xGTO) 1þe 6:9115þ(0:03912xGTO) [Equation 2]

In summary, this is an excellent manuscript which provides a pathway for future studies which would allow for the more rapid determination of futility for severely bleeding trauma patients during a period of urgent need to conserve blood products through improved hemovigilance and blood component stewardship.

Author Response

Comment 1: The author’s manuscript was most likely written and submitted during the latter part of 2024 before which a few important papers had been published which support the authors conclusions. Two of these papers I have co-authors, and the authors are not allowed to cite these papers. Specifically, Al Fadhl et al., Walsh et al. and Bhogadi et al. have clearly described early markers for defining futility for severely bleeding trauma patients who also have associated head injury. In those three papers, the presence of severe brain injury markedly worsens the prognosis such that the presence of associated severe TBI is an independent factor which weighs heavily on the decision to terminate futile renunciation early in the course of trauma resuscitation for severely bleeding trauma with severe TBI and extracranial injury. The Bhogadi et al. paper may be cited.

These papers are noted below.

Al-Fadhl MD, Karam MN, Chen J, Zackariya SK, Lain MC, Bales JR, Higgins AB, Laing JT, Wang HS, Andrews MG, Thomas AV, Smith L, Fox MD, Zackariya SK, Thomas SJ, Tincher AM, Al-Fadhl HD, Weston M, Marsh PL, Khan HA, Thomas EJ, Miller JB, Bailey JA, Koenig JJ, Waxman DA, Srikureja D, Fulkerson DH, Fox S, Bingaman G, Zimmer DF, Thompson MA, Bunch CM, Walsh MM; Futile Indicators for Stopping Transfusion in Trauma (FISTT) Collaborative Group. Traumatic Brain Injury as an Independent Predictor of Futility in the Early Resuscitation of Patients in Hemorrhagic Shock. J Clin Med. 2024 Jul 3;13(13):3915. doi: 10.3390/jcm13133915. PMID: 38999481; PMCID: PMC11242176.

Walsh MM, Fox MD, Moore EE, Johnson JL, Bunch CM, Miller JB, Lopez-Plaza I, Brancamp RL, Waxman DA, Thomas SG, Fulkerson DH, Thomas EJ, Khan HA, Zackariya SK, Al-Fadhl MD, Zackariya SK, Thomas SJ, Aboukhaled MW, The Futile Indicators For Stopping Transfusion In Trauma FISTT Collaborative Group. Markers of Futile Resuscitation in Traumatic Hemorrhage: A Review of the Evidence and a Proposal for Futility Time-Outs during Massive Transfusion. J Clin Med. 2024 Aug 9;13(16):4684. doi: 10.3390/jcm13164684. PMID: 39200824; PMCID: PMC11355875.

Bhogadi SK, Ditillo M, Khurshid MH, Stewart C, Hejazi O, Spencer AL, Anand T, Nelson A, Magnotti LJ, Joseph B. Development and Validation of Futility of Resuscitation Measure in Older Adult Trauma Patients. J Surg Res. 2024 Sep;301:591-598. doi: 10.1016/j.jss.2024.07.019. Epub 2024 Aug 1. PMID: 39094517. ( This paper may be cited since I am not a coauthor. The senior author of this paper,  Bellal Joseph, is a leading proponent of the GTOS and he is searching for new bedside predictors to refine the GTOS to identify futility in trauma resuscitation much earlier in order to preserve scarce resources.

Response 1: We appreciate the reviewer’s insightful feedback and for highlighting these relevant studies. We recognize their importance in reinforcing the concept that severe TBI serves as an independent predictor of futility in trauma resuscitation. Additionally, we strongly agree that the early identification of patients unlikely to benefit from resuscitative efforts can help optimize clinical decision-making and resource allocation. We sincerely thank the reviewer for guiding us in strengthening our manuscript with these valuable insights.

Comment 2: A motivation for the above mentioned papers is that there are no current guidelines which provide clear direction for defining futility for severely bleeding trauma patients with or without associated TBI. The need for such guidelines which can be applied earlier in the course of trauma resuscitation has been brought to the forefront by the dwindling supply of blood products available for trauma resuscitation for these patients. The cause of the significant blood shortage in the United states, for example, is, among other causes, the increased use of hemostatic resuscitation whereby, patents are resuscitated with large quantities of blood products in ratios of 1/1/1/ packed red cells, plasma and platelets with increasing use of whole blood. The unintended consequence of this strategy is a shortage blood products for future trauma resuscitation as well as for other emergencies. Recent literature has documented the great strain in blood banks which are now faced with shortages of vital blood components during peak periods of trauma.

I would therefore suggest that the authors add the paper by described above. The first two papers cannot be cited.

Bhogadi SK, Ditillo M, Khurshid MH, Stewart C, Hejazi O, Spencer AL, Anand T, Nelson A, Magnotti LJ, Joseph B. Development and Validation of Futility of Resuscitation Measure in Older Adult Trauma Patients. J Surg Res. 2024 Sep;301:591-598. doi: 10.1016/j.jss.2024.07.019. Epub 2024 Aug 1. PMID: 39094517. ( This paper may be cited since I am not a coauthor. The senior author of this paper,  Bellal Joseph, is a leading proponent of the GTOS and he is searching for new bedside predictors to refine the GTOS to identify futility in trauma resuscitation much earlier in order to preserve scarce resources.

to their bibliography and also highlight the importance of using the presence of TBI as an important marker for futility which, therefore, will assist traumatologists in identifying those patients with extracranial hemorrhage who will not survive resuscitative efforts.

Response 2: The need for clear guidelines to define futility in severely bleeding trauma patients, particularly those with associated TBI, remains a critical issue. The absence of standardized criteria for guiding early decision-making in trauma resuscitation for these patients creates a significant gap in clinical practice. Furthermore, we strongly agree that the ongoing shortage of blood products is becoming an increasingly pressing concern. The reviewer’s insights on this issue have been highly valuable in refining our discussion. Therefore, we have fully incorporated the reviewer's valuable suggestions into the manuscript in accordance with the agreements made in Response 1 and 2. The recommended references have been added to our bibliography, and the Discussion section has been revised to highlight the significance of severe TBI as a key marker of futility in trauma resuscitation. This content has been added to the Discussion section, lines 237–249. This insightful recommendation has enhanced the clinical relevance of our study.

Comment 3: In addition, a passing reference to the need to more clearly define mortality early during trauma resuscitation because of the shortage of blood products would make this paper more relevant to the traumatologist who will need to use tools like the GTOS to predict death earlier during the course of trauma resuscitation in order to spare blood products for those younger patients without TBI who will benefit from continued resuscitation. These papers are the following.

Van Gent JM, Clements TW, Lubkin DT, Wade CE, Cardenas JC, Kao LS, Cotton BA. Predicting Futility in Severely Injured Patients: Using Arrival Lab Values and Physiology to Support Evidence-Based Resource Stewardship. J Am Coll Surg. 2023 Apr 1;236(4):874-880. doi: 10.1097/XCS.0000000000000563. Epub 2023 Jan 18. PMID: 36728085.

Clements TW, Van Gent JM, Lubkin DE, Wandling MW, Meyer DE, Moore LJ, Cotton BA. The reports of my death are greatly exaggerated: An evaluation of futility cut points in massive transfusion. J Trauma Acute Care Surg. 2023 Nov 1;95(5):685-690. doi: 10.1097/TA.0000000000003980. Epub 2023 May 1. PMID: 37125814.

Loudon AM, Rushing AP, Hue JJ, Ziemak A, Sarode AL, Moorman ML. When is enough enough? Odds of survival by unit transfused. J Trauma Acute Care Surg. 2023 Feb 1;94(2):205-211. doi: 10.1097/TA.0000000000003835. Epub 2022 Nov 10. PMID: 36694331.

Saillant NN, Kornblith LZ, Moore H, Barrett C, Schreiber MA, Cotton BA, Neal MD, Makar R, Cap AP. The National Blood Shortage-An Impetus for Change. Ann Surg. 2022 Apr 1;275(4):641-643. doi: 10.1097/SLA.0000000000005393. PMID: 35081570; PMCID: PMC9055632.

Mladinov D, Frank SM. Massive transfusion and severe blood shortages: establishing and implementing predictors of futility. Br J Anaesth. 2022 Feb;128(2):e71-e74. doi: 10.1016/j.bja.2021.10.013. Epub 2021 Nov 15. PMID: 34794769.

Response 3: Thank you for your insightful suggestion. In response to your comment, we have incorporated a reference to the need for clearly defining early mortality during trauma resuscitation, particularly in the context of blood product shortages. We acknowledge the importance of utilizing predictive tools like the GTOS to facilitate early decision-making in trauma resuscitation, thereby optimizing the allocation of limited blood resources to patients who are most likely to benefit, such as younger patients without TBI. As per your recommendation, we have added this discussion in lines 245–247 of the manuscript.

Comment 4: Again, as mentioned above; in order to display transparency, this reviewer is a coauthor for the first two papers. The authors may not use these two papers because of the potential conflict of interest since I have suggested these papers. However, these two papers represent the most complete reviews regarding the effect of TBI on the likelihood of survival for trauma patients in severe shock who are receiving large quantities blood products. There are other less current and comprehensive papers which address the same concerns which I will add below.

Huang CY, Yen YH, Tsai CH, Hsu SY, Tsai PL, Hsieh CH. Geriatric Trauma Outcome Score as a Mortality Predictor in Isolated Moderate to Severe Traumatic Brain Injury: A Single-Center Retrospective Study. Healthcare (Basel). 2024 Aug 22;12(16):1680. doi: 10.3390/healthcare12161680. PMID: 39201238; PMCID: PMC11353928. (Very recent publication and cited as reference 17 already.)

Saillant NN, Kornblith LZ, Moore H, Barrett C, Schreiber MA, Cotton BA, Neal MD, Makar R, Cap AP. The National Blood Shortage-An Impetus for Change. Ann Surg. 2022 Apr 1;275(4):641-643. doi: 10.1097/SLA.0000000000005393. PMID: 35081570; PMCID: PMC9055632.

L'Huillier JC, Hua S, Logghe HJ, Yu J, Myneni AA, Noyes K, Guo WA. Transfusion futility thresholds and mortality in geriatric trauma: Does frailty matter? Am J Surg. 2024 Feb;228:113-121. doi: 10.1016/j.amjsurg.2023.08.020. Epub 2023 Aug 25. PMID: 37684168.

Mladinov D, Frank SM. Massive transfusion and severe blood shortages: establishing and implementing predictors of futility. Br J Anaesth. 2022 Feb;128(2):e71-e74. doi: 10.1016/j.bja.2021.10.013. Epub 2021 Nov 15. PMID: 34794769.

Maegele M, Schöchl H, Menovsky T, Maréchal H, Marklund N, Buki A, Stanworth S. Coagulopathy and haemorrhagic progression in traumatic brain injury: advances in mechanisms, diagnosis, and management. Lancet Neurol. 2017 Aug;16(8):630-647. doi: 10.1016/S1474-4422(17)30197-7. Epub 2017 Jul 11. PMID: 28721927.

Park J, Lee Y. Predicting Mortality of Korean Geriatric Trauma Patients: A Comparison between Geriatric Trauma Outcome Score and Trauma and Injury Severity Score. Yonsei Med J. 2022 Jan;63(1):88-94. doi: 10.3349/ymj.2022.63.1.88. PMID: 34913288; PMCID: PMC8688368. ( already cited as reference 13)

Benhamed A, Emond M, Mercier E, Heidet M, Gauss T, Saint-Supery P, Yadav K, David JS, Claustre C, Tazarourte K. Accuracy of a Prehospital Triage Protocol in Predicting In-Hospital Mortality and Severe Trauma Cases among Older Adults. Int J Environ Res Public Health. 2023 Jan 20;20(3):1975. doi: 10.3390/ijerph20031975. PMID: 36767343; PMCID: PMC9916137.

Raj R, Wennervirta JM, Tjerkaski J, Luoto TM, Posti JP, Nelson DW, Takala R, Bendel S, Thelin EP, Luostarinen T, Korja M. Dynamic prediction of mortality after traumatic brain injury using a machine learning algorithm. NPJ Digit Med. 2022 Jul 18;5(1):96. doi: 10.1038/s41746-022-00652-3. PMID: 35851612; PMCID: PMC9293936.

Zakrison TL, Essig R, Polcari A, McKinley W, Arnold D, Beyene R, Wilson K, Rogers S Jr, Matthews JB, Millis JM, Angelos P, O'Connor M, Mansour A, Goldenberg F, Spiegel T, Horowitz P, Das P, Slidell M, Chokshi N, Okeke I, Barth R, Wilkins HE 3rd, Kass-Hout T, Lazaridis C. Review Paper on Penetrating Brain Injury: Ethical Quandaries in the Trauma Bay and Beyond. Ann Surg. 2023 Jan 1;277(1):66-72. doi: 10.1097/SLA.0000000000005608. Epub 2022 Aug 23. PMID: 35997268; PMCID: PMC9762724.

Response 4: As previously mentioned, we have accepted the reviewer’s suggestion. We believe that incorporating these studies strengthens the scientific foundation of our manuscript. Additionally, we acknowledge the other references provided and have reviewed them for additional context. Thank you for the thoughtful recommendation and for contributing to the rigor of our study.

Comment 5: Finally, the authors state on pages 6 and 7 lines 168-173 that: “The GTOS has been widely recognized as a practical tool for predicting in-hospital mortality 167 in geriatric trauma patients, integrating three simple yet clinically relevant factors: age, 168 ISS, and the need for RBC transfusion within the first 24 h.” The reason for my enthusiasm for this manuscript is that the paper confirms our position and that of others recently in the literature as mentioned above that more accurate indicators of early death are needed to better predict those patient who will certainly die during resuscitation of severely bleeding trauma patients in order to preserve scare blood components. This paper correctly states that the severity of TBI in elderly trauma patients is an independent predictor of mortality. However, the calculation of the GTOS is not “simple” mentioned by the authors and cannot be dome at the bedside in the Emergency Department. In the paper cited above. The authors cannot cite the Al Fadhl paper, but I am copying a section from that paper which clearly analyzes why the GTOS, and the frailty index are not suitable toll of the early identification of the bleeding trauma patient who will not survive the first hours of resuscitation which is when a large preponderance of blood products and resources are used. I believe that the section in italics below will help the authors refine their message to refer to the urgent need for prediction toll which can predicts early death in bleeding trauma patients with head injury.

Al-Fadhl MD, Karam MN, Chen J, Zackariya SK, Lain MC, Bales JR, Higgins AB, Laing JT, Wang HS, Andrews MG, Thomas AV, Smith L, Fox MD, Zackariya SK, Thomas SJ, Tincher AM, Al-Fadhl HD, Weston M, Marsh PL, Khan HA, Thomas EJ, Miller JB, Bailey JA, Koenig JJ, Waxman DA, Srikureja D, Fulkerson DH, Fox S, Bingaman G, Zimmer DF, Thompson MA, Bunch CM, Walsh MM; Futile Indicators for Stopping Transfusion in Trauma (FISTT) Collaborative Group. Traumatic Brain Injury as an Independent Predictor of Futility in the Early Resuscitation of Patients in Hemorrhagic Shock. J Clin Med. 2024 Jul 3;13(13):3915. doi: 10.3390/jcm13133915. PMID: 38999481; PMCID: PMC11242176. state:  “The parameter of age is considered particularly significant in the foundation studies that define the severity of TBI, although the effect of the increasing severity of frailty on the outcomes is more important and can be added to the list of predictors of mortality for the geriatric trauma patient, such as the Geriatric Trauma Outcome Score (GTOS) and the Brain Injury Guidelines (BIG), which require the calculation of a post-admission Injury Severity Score (ISS) [71–81]. In the most recent studies to identify predictors of FR in patients with TBI, the accurate scoring of a “frailty index” in trauma patients can prevent unnecessary interventions in those with a high risk of dying while not withholding intervention in those who may find it beneficial [79,80]. However, the frailty index requires the calculation of parameters that cannot be calculated at the bedside within the first few hours of arrival; therefore, there is an impetus to find triage tools for patients with TBI that can predict early mortality [32,74,76,79,82–88].

 The GTOS is not simple and cannot be easily calculated at the bedside in the ED however, this paper is an important step toward the development of bedside criteria which can accurately predict futility in severely bleeding trauma patients.

The formula for calculation of the GTOS is noted below to confirm that the score is not always simple and cannot be calculated at the bedside within the first few hours of resuscitation. This point is being made because the search for more accurate indicators of certain death for severely bleeding trauma patients will include the assessment of age and the presence of severe TBI as the first marker of futility during the earliest periods resuscitation ad will rely heavily on this type of research that the authors have skillfully presented in their submitted manuscript to Diagnostics.

 The calculation involves two equations in order to show predictive value as noted by:

Zhao FZ, Wolf SE, Nakonezny PA, Minhajuddin A, Rhodes RL, Paulk ME, Phelan HA. Estimating Geriatric Mortality after Injury Using Age, Injury Severity, and Performance of a Transfusion: The Geriatric Trauma Outcome Score. J Palliat Med. 2015 Aug;18(8):677-81. doi: 10.1089/jpm.2015.0027. Epub 2015 May 14. PMID: 25974408; PMCID: PMC4522950.

GTO score¼ Age þ (2:5 x ISS)þ22 (if given PRBCs) [Equation 1]

GTO logistic model¼ e 6:9115þ(0:03912xGTO) 1þe 6:9115þ(0:03912xGTO) [Equation 2]

In summary, this is an excellent manuscript which provides a pathway for future studies which would allow for the more rapid determination of futility for severely bleeding trauma patients during a period of urgent need to conserve blood products through improved hemovigilance and blood component stewardship.

Response 6: Thank you for your precise observation. I completely agree with the reviewer’s opinion that GTOS is not a simplistic measure. I have revised this section in the Discussion, lines 172–173.

Reviewer 3 Report

Comments and Suggestions for Authors

The authors conducted a retrospective analysis of elderly patients with TBI. Although it was a single-center study,  it incorporated many subjects over a long period. This study showed the inconsistency of GTOS in predicting outcomes in older patients. Overall, I think this study was well written, well conducted, and fit for publication. I only have several very minor concerns. 

  1. Please check the spelling and grammar (for example, line 85)
  2. Fall is said to be the leading cause of TBI in the elderly. The authors differed fall and split in Table 1. I wonder what the difference was.

Congratulation for the excellent work. 

Comments on the Quality of English Language

I don't have any capacity to assess the quality of the English language, but I don't have any issues understanding this manuscript. 

Author Response

Comment 1: Please check the spelling and grammar (for example, line 85)

Response 1: We sincerely appreciate you pointing out our error. We have thoroughly reviewed the entire manuscript and have specifically corrected the section you highlighted.

Comment 2: Fall is said to be the leading cause of TBI in the elderly. The authors differed fall and split in Table 1. I wonder what the difference was.

Response 2: I would like to clarify whether you were referring to "fall and slip" rather than "fall and split." Assuming that my understanding is correct, I will provide my response accordingly.

In the Korean Trauma Data Base (KTDB) used in this study, the injury mechanism distinguishes between "fall" and "slip down." Slip down is defined as an injury resulting from tripping or slipping at ground level. In contrast, fall refers to a situation where the body drops from a certain height. Specifically, a fall from a height of 6 meters or more in adults and 3 meters or more in children is considered a severe injury mechanism.

Thank you for your detailed and thoughtful question.